# miRNA Changes in Retinal Ganglion Cells after Optic Nerve Crush and Glaucomatous Damage

**DOI:** 10.3390/cells10071564

**Published:** 2021-06-22

**Authors:** Ben Mead, Alicia Kerr, Naoki Nakaya, Stanislav I. Tomarev

**Affiliations:** 1School of Optometry and Vision Sciences, Cardiff University, Cardiff CF24 4HQ, UK; 2Section of Retinal Ganglion Cell Biology, Laboratory of Retinal Cell and Molecular Biology, National Eye Institute, National Institutes of Health, Bethesda, MD 20892, USA; alicia.kerr@nih.gov (A.K.); nakayan@nei.nih.gov (N.N.)

**Keywords:** retinal ganglion cells, miRNA, glaucoma, optic neuropathy

## Abstract

The purpose of this study was to characterize the miRNA profile of purified retinal ganglion cells (RGC) from healthy and diseased rat retina. Diseased retina includes those after a traumatic optic nerve crush (ONC), and after ocular hypertension/glaucoma. Rats were separated into four groups: healthy/intact, 7 days after laser-induced ocular hypertension, 2 days after traumatic ONC, and 7 days after ONC. RGC were purified from rat retina using microbeads conjugated to CD90.1/Thy1. RNA were sequenced using Next Generation Sequencing. Over 100 miRNA were identified that were significantly different in diseased retina compared to healthy retina. Considerable differences were seen in the miRNA expression of RGC 7 days after ONC, whereas after 2 days, few changes were seen. The miRNA profiles of RGC 7 days after ONC and 7 days after ocular hypertension were similar, but discrete miRNA differences were still seen. Candidate mRNA showing different levels of expression after retinal injury were manipulated in RGC cultures using mimics/AntagomiRs. Of the five candidate miRNA identified and subsequently tested for therapeutic efficacy, miR-194 inhibitor and miR-664-2 inhibitor elicited significant RGC neuroprotection, whereas miR-181a mimic and miR-181d-5p mimic elicited significant RGC neuritogenesis.

## 1. Introduction

Retinal ganglion cells (RGC) are the sole projection neurons of the retina to the brain, and their axons makes up the optic nerve. Alongside the brain and spinal cord, the retina makes up the central nervous system (CNS). A key characteristic of the CNS is its inability to regenerate damaged axons, or replace lost neurons, making any injury particular devastating and the resulting functional deficits often permanent. From a clinical perspective, they are not only the one of the cell types most responsible for irreversible vision loss (e.g., glaucoma and traumatic optic neuropathy), but also an easily accessible model used to study spinal cord injury and, to a lesser extent, traumatic brain injury [1]. It is for this reason that RGC have become perhaps one of the most attractive cell types from a research perspective. The optic nerve is the main site of damage, which can come about due to trauma or degeneration.

Trauma, often referred to as traumatic optic neuropathy (TON), is an acute sundering of the optic nerve and the RGC axons within. The consequences of such an injury are analogous to a spinal cord injury; the axons undergo Wallerian degeneration, attempting, but ultimately failing to regenerate. An immediate loss of function (vision) occurs, and never returns. Distinct from spinal cord injury, however, the RGC degenerate. CNS neurons depend on their connections to retrogradely supply survival-inducing neurotrophic factors (NTF). In the spinal cord, a severed axon does not cut off this supply as lateral connections to nearby interneurons are still preserved. In RGC, a TON, modeled using an optic nerve crush (ONC), removes 100% of the RGC’s supply of NTF, and cell loss shortly follows. Approximately 50% of RGC are lost 7 days post TON, and 90% after 2–3 weeks [2]. Axotomy also induces glial changes, such as a shift in the astrocyte phenotype to the neurotoxic astrocyte, which further mediates the neurodegenerative process [3].

The most common injury RGC succumb to is not TON, but glaucoma. Glaucoma is characterized by the slow progressive degeneration of RGC with principal risk factors including intraocular pressure (IOP) and age. Glaucoma affects 80 million people [4] and treatment is restricted to the management of its risk factor, IOP. The mechanisms by which ocular hypertension kills RGC are unknown, although many theories have been proposed [5,6,7].

From these theories, a variety of neuroprotective agents have been discovered and their therapeutic efficacy is often effective both in TON and glaucoma although these two pathologies are clinically different. For example, BAX knockout is effective after TON [8] and glaucoma [9], as are the use of stem cells [10] and extracellular vesicles/exosomes [11].

Understanding the processes by which RGC degenerate and die is imperative to the development of new treatments. Several studies have characterized the mRNA profile of RGC before and after injury. These data led to the identification of several signaling pathways involved in the degeneration process [12,13]. While previous studies have determined changes in the miRNA profile of total retina or aqueous humor in health and disease [14,15], none, to our knowledge, have focused on the miRNA profile of RGC specifically.

miRNA are non-coding RNA whose function is to silence mRNA, preventing protein translation. Mature miRNA are 22 nucleotides in length and guide argonaute proteins to mRNA whose sequence is complementary to that of the miRNA [16]. Since the miRNA only needs as few as seven nucleotides that are complementary to the mRNA sequence [17], each miRNA may bind to 100 s or even 1000 s of different mRNA. Once the miRNA bind to the complementary mRNA, translational efficiency is inhibited and mRNA may even be degraded entirely. The roles of miRNA are poorly understood; however, evidence suggests that their effects are not to induce significant changes in cellular function but, rather, to maintain homeostasis and protect against fluctuations in gene expression as well as direct cell fate and developmental timing [18,19,20].

The following study sought to understand the miRNA profile of purified RGC, both in healthy retina and after TON and glaucoma/ocular hypertension. Using these miRNA profiles, we then tested candidate therapies designed for modifying/correcting this miRNA profile.

## 2. Materials and Methods

### 2.1. Animals

Adult female outbred Sprague-Dawley rats weighing 150 to 200 g (Charles River, Wilmington, MA, USA) and adult 2–3-month-old C57BL/6J mice (the Jackson Laboratory, Bar Harbor, ME, USA) were maintained in accordance with guidelines described in the ARVO Statement for the Use of Animals in Ophthalmic and Vision Research, using protocols approved by the National Eye Institute Committee on the Use and Care of Animals/Cardiff University’s Biological Standard’s committee (NEI-556: “Retinal ganglion cell neuroprotection and axon regeneration in mouse glaucoma models and after optic nerve crush”. Approved 21 August 2018. Active for 3 years. NEI-586: “Stem cell therapy for glaucoma and optic nerve crush”. Approved 21 June 2019. Active for 3 years). Animals were kept at 21 °C and 55% humidity under a 12-hour light and dark cycle, given food/water ad libitum and were under constant supervision from trained staff. Animals were euthanized by rising concentrations of CO_2_ before extraction of retinae.

### 2.2. Materials

All reagents were purchased from Sigma (Allentown, PA, USA) unless otherwise specified.

### 2.3. In Vivo Experimental Design

The experimental design is shown schematically in Figure 1. Rats were divided into four groups: Group 1 consisted of three uninjured/untreated animals; Group 2 consisted of three rats seven days after ocular hypertension was induced by laser photocoagulation of the trabecular meshwork (TM) and limbal vessels; Group 3 consisted of three rats two days post-ONC; and Group 4 consisted of three rats 7 days post-ONC.

To make up for any animals/ tissue that were unusable at the end of the study (due to animal deaths, surgical problems, lack of ocular hypertension), additional animals were run to ensure each treatment subgroup was made up of three animals/six eyes.

### 2.4. Induction of Ocular Hypertension with Laser Photocoagulation (Rats)

Ocular hypertension was induced in Group 2 by laser photocoagulation of the TM and circumferential limbal vessels as previously described [21,22]. Anaesthesia was induced with intraperitoneal injection of Ketamine (100 mg/kg; Putney Inc, Portland, ME, USA)/Xylazine (10 mg/kg; Lloyd Inc, Shenandoah, IA, USA). Pupil constriction and subsequent opening of the iridocorneal angle was achieved with 4% pilocarpine hydrochloride ophthalmic solution (Sandoz, Princeton, NJ, USA). An OcuLight GLx 532 nm laser (Iridex, Mountain View, CA, USA) was used to deliver laser burns at 0.3 W, at a spot size of 100 µm and duration of 0.5 s. Three locations were photocoagulated: approximately 270° of the circumferential limbal vessels, episcleral veins branching from these limbal vessels and, finally, a trans-scleral/trans-corneal 360° burn of the TM/iridocorneal angle. Nasal vasculature was left uninjured to prevent ischemia.

### 2.5. Induction of Ocular Hypertension with Silicon Oil Injection (Mice)

Ocular hypertension in mice was induced by silicone oil injection into the anterior chamber of the eye, as previously described [23]. Mice were anesthetized by an intraperitoneal injection of Avertin (0.3 mg/g). A 32G needle was used to tunnel through the layers of the cornea to reach the anterior chamber without injuring the lens or iris. Following this entry, about 2 µL silicone oil (Silikon, Alcon Laboratories, Fort Worth, TX, USA) was injected slowly into the anterior chamber using a sterile glass micropipette.

### 2.6. Intraocular Pressure Recording

IOP were recorded for all rats and mice using a Tonolab rebound tonometer (Colonial Medical Supply, Franconia, NH, USA). IOP was recorded under isoflurane-induced anesthesia for rats during the same 3-hour window each day, sampled 18 times and averaged for each individual recording. Rats were considered “7 days post-glaucoma” at day 7 whereby day 0 was the first day of a measurable elevated pressure recording. IOP in mice was measured under isoflurane-induced anesthesia, one week post injection of silicone oil. Mice were placed in a chamber with isoflurane and oxygen and, after 15 min of anesthesia, IOP was measured. Careful note was taken to make sure that cornea did not dry out in that time. An elevated IOP was defined at 40% above baseline, typically from 10 mmHg to 14 mmHg (data not shown).

### 2.7. Optic Nerve Crush

TON was induced in Group 3 and 4 by surgically-induced ONC. Anesthesia was induced with 5% Isoflurane (Baxter Healthcare Corp, Deerfield, IL, USA)/1.5 L per minute O_2_ and maintained at 3.5% throughout the procedure. Following anesthetic induction, an intraperitoneal injection of Buprenorphine (0.3 mg/kg) was administered (preoperatively) and the animal secured in a heal-holding frame. Intraorbital ONC was performed as previously described [24]. Briefly, the optic nerve was surgically exposed under the superior orbital margin and crushed using fine forceps 1 mm posterior to the lamina cribrosa, taking care to separate the dura mater and underlying retinal artery before crushing.

### 2.8. Isolation, Purification, and/or Culture of Retinal Ganglion Cells

Eight well chamber slides (Thermo Fisher Scientific, Waltham, MA, USA) were pre-coated with 100 µg/mL poly-D-lysine for 60 min and then with 20 µg/mL laminin for 30 min. After culling and ocular dissection, the retinae of female Sprague-Dawley were minced in 1.25 mL of papain (20 U/mL; Worthington Biochem, Lakewood, NJ, USA; as per manufacturer’s instructions (#LK003150)) containing 50 µg/mL of DNase I (62.5 µL; Worthington Biochem) and incubated for 90 min at 37 °C. The retinal cell suspension was centrifuged at 300× *g* for 5 min and the pellet resuspended in 1.575 mL of Earle’s balanced salt solution (Worthington Biochem) containing 1.1 mg/mL of reconstituted albumin ovomucoid inhibitor (150 µL; Worthington Biochem) and 56 µg/mL of DNase I (75 µL). After adding to the top of 2.5 mL of albumin ovomucoid inhibitor (10 mg/mL) to form a discontinuous density gradient, the retinal cell suspension was centrifuged at 70× *g* for 6 min and the cell pellet resuspended in 1 mL of PBS. Retinal cell suspensions then underwent one of two distinct methodologies. For miRNA sequencing, the cell suspensions went through further purification to obtain RGC. For in vitro testing of the neuroprotective/regenerative capacity of miRNA mimic/inhibitors, the cell suspension was grown as a heterogenous retinal culture.

For heterogenous retinal cultures, retinal cells were plated as a density of 125,000 cells per well in supplemented Neurobasal-A (25 mL Neurobasal-A (Thermo Fisher Scientific), 1× concentration of B27 supplement (Life Technologies, Carlsbad, CA, USA), 0.5 mM of L-glutamine (62.5 µL; Thermo Fisher Scientific) and 50 µg/mL of gentamycin (125 µL; Thermo Fisher Scientific)).

For miRNA sequencing, RGC were purified from the retinal suspension using CD90.1 (Thy1) magnetic beads as per the manufacturer’s instructions (Miltenyi Biotec, Auburn, CA, USA; #130-096-209). Briefly, retinal cells are incubated with CD90.1 enrichment and CD11b depletion antibodies conjugated to magnetic beads. Following depletion, the retinal suspension is passed through a magnetised column and the enriched RGC are collected. Since there is a possibility that some Thy1^+^ amacrine subtypes will have also been included in the purification, a portion of RGC were plated at a density 5000 RGC/well to confirm RGC purity. We confirmed >99% RGC purity by immunocytochemistry, staining for RBPMS (Appendix A).

### 2.9. RNA Sequencing

RNAseq was performed by LC Sciences on RNA isolated from purified RGC. RNA was isolated using Trizol/Qiagen RNeasy MaxiPrep kit and quantified using Agilent Bioanalyzer (Agilent, Santa Clara, CA, USA). Libraries were constructed and sequenced using Illumina NextSeq instrument with 1 × 75 bp single-end reads at an approximate depth of 10–15 million reads per sample. A scaling factor for a given sample was computed as the median of the ratio of its read count for each gene over its geometric mean across all samples. Raw read counts were divided by the factor associated with their samples for normalization. Unlike protein-coding genes/mRNA-seq data analysis, in which only uniquely mapped reads are considered, the miRNA pipeline needs to allow multiple mapping of the same read to account for the multiple copies. Thus, normalization was conducted on the number of read alignments mapped to annotated gene features across samples instead of the number of mapped reads.

The RNAseq data was displayed as a heat map of the log_2_ fold change of RGC from Group 2, 3 and 4, in comparison to healthy RGC (Group 1). A miRNA was considered differentially abundant when the log_2_ fold change was >2 or <−2.

### 2.10. In Situ Hybridization-Mediated Localization of miRNA in Mouse Retina

C57BL/6J mice aged 3 months were euthanized via rising CO_2_ levels. Retinas were dissected and immediately frozen with dry ice. In situ hybridization was performed on 14 µm sections. 

miRNAscope HD Assay Red (Advanced Cell Diagnostics, La Jolla, CA, USA: Cat #324500) was used to visualize miRNA according to ACD Fresh Frozen tissue protocol. Slides were imaged using Zeiss AxioImager Z1 (Carl Zeiss, Oberkochen, Germany).

### 2.11. miRNA Modulation Treatment of Heterogeneous Retinal Cultures

Retinal cultures were treated 6 h after cells were plated and, on day 3, were stained via immunocytochemistry and analysed for neuroprotective and neuritogenic effects. Five miRNA were selected that were showing the greatest deviations from healthy retina. For the miRNA that were reduced after injury (miR-181a-3p and miR-181d-5p), miRNA mimics were delivered into retinal cultures, whereas for those miRNA that were increased (miR-194-5p, miR-708-5p, and miR-664-2-5p), mirVana miRNA inhibitors were added. Transfection was achieved using lipofectamine RNAiMAX according to the manufacturer’s instructions. Briefly, 30 nM of the miRNA was premixed with 3 µL of lipofectamine for 15 min before addition into the well.

### 2.12. Immunocytochemistry and Microscopy

Heterogenous retinal cells were fixed in 4% paraformaldehyde (PFA) in phosphate-buffered saline (PBS) for 10 min, washed for 3 × 10 min of PBS, blocked in blocking solution (3% bovine serum albumin (g/mL), 0.1% Triton ×(−100) in PBS) for 20 min and incubated with mouse anti rat βIII-tubulin antibody diluted at 1:500 in antibody diluting buffer (ADB; 0.5% bovine serum albumin, 0.3% Tween-20 in PBS) for 1 h at room temperature. Cells were then washed for 3 × 10 min in PBS, incubated with the goat anti mouse AlexaFluor488 antibody diluted at 1:400 in ADB for 1 h at room temperature, washed for 3 × 10 min in PBS, mounted in Vectorshield mounting medium containing DAPI (Vector Laboratories, Peterborough, UK) and stored at 4 °C. Retinal cultures were imaged using a Leica DM6000B/AF6000 fluorescence imaging system (Leica, Wetzlar, Germany). The entire 8 well chamber was imaged and the number of βIII-tubulin^+^ RGC/RGC with neurites was quantified. Counts were conducted manually by an individual masked to the treatment groups. All cell cultures were performed both in triplicate, and three separate times (*n* = 3).

### 2.13. Statistics

All statistical tests were performed using SPSS 17.0 (IBM SPSS, Inc., Chicago, IL, USA) and graphs constructed using Graphpad Prism 7.01 (Graphpad Prism, La Jolla, CA, USA) and Microsoft Excel (Microsoft, Redmond, WA, USA). The Shapiro–Wilk test was used to ensure all data were normally distributed before parametric testing using a 1-way ANOVA with a Tukey post hoc test. Statistical differences were considered significant at *p* values < 0.05.

## 3. Results

### 3.1. Ocular Hypertension and TON Induce Significant Changes in RGC miRNA

We identified over 100 miRNA whose abundance was significantly (*p* < 0.001) different in injured RGC (Group 2, 3, and 4) in comparison to uninjured RGC (Group 1; Figure 2). The total estimated abundance counts for each miRNA are shown in Figure 3. Among the 30 most abundant miRNA in Group 1, only two (mir-7a-5p and mir-191a-5p) were not identified among the most abundant miRNA in several analyzed neuronal cell types and retinal Muller glial cells [25,26,27] (Table 1).

Of note, several miRNA differences were identified between the injured RGC groups. For example, between Group 3 and 4, of which the difference is 2 and 7 days after TON (respectively), many miRNA do not significantly change until after 7 days (Group 4). One of the most significant examples is miR-194a-5, which shows no changes 2 days after TON but an over 64-fold increase 7 days after TON. A similar observation is seen with miR-664-2-5p and miR-708-5p, albeit to a lesser extent. Between Group 2 and 4, which compares TON to glaucoma, differences are still present but less pronounced. For example, the miR-181 family are decreased significantly in glaucoma (Group 2) but are unchanged or decrease less significantly in Group 4 (TON).

### 3.2. Changes in miR-181d-5p Level in the Retina after IOP Elevation

To verify changes in the level of certain miRNA in the ganglion cell layer (GCL) after elevation of IOP, we used in situ hybridization. miR-181d-5p, that was reduced after injury according to sequencing data, was used for these experiments. Let-7b-5p was used as a control as it remains unchanged after ocular hypertension (data not shown). Both miRNA are expressed in the GCL and inner nuclear layer (INL) in control retina (Figure 4A,C). The level of let-7b-5p did not change much after IOP elevation, while the level of miR-181d-5p was significantly reduced in the GCL and somewhat reduced in the INL (Figure 4D).

### 3.3. Modulation of Candidate miRNA in RGC Promotes Neuroprotection and Neuritogenesis

Candidate miRNA were selected as those that had the most substantial differences from healthy intact retina. Treatment of RGC cultures with miR-664-2-5p or miR-194-5p inhibitors elicited significant survival (Figure 5) of RGC (252 ± 19 and 230.3 ± 26.5 RGC/well, respectively) compared to both untreated RGC (114 ± 4 RGC/well) and lipofectamine-treated controls (84.3 ± 4.3 RGC/well). Treatment with miR-708-5p inhibitor, miR-181d-5p mimic, and miR-181a-3p mimic elicited no significant neuroprotective effect (86.3 ± 5.9, 97.3 ± 7, and 94.7 ± 3.5 RGC/well, respectively) compared to untreated/lipofectamine-treated controls.

Neuritogenesis was measured as the number of neurite-bearing RGC (Figure 5). Treatment of RGC cultures with miR-181d-5p or miR-181a-3p mimics elicited significant neuritogenesis of RGC (124.7 ± 10.5 and 112.3 ± 8.1 RGC with neurites, respectively) compared to both untreated RGC (20.7 ± 4.6 RGC with neurites) and lipofectamine-treated controls (12.7 ± 4.3 RGC with neurites). Treatment with miR-664-2-5p or miR-194-5p inhibitors elicited a trend towards neuritogenesis (40.7 ± 4.9 and 38.7 ± 4.8 RGC with neuritis, respectively) but did not reach significance (*p*-value of 0.056 and 0.054, respectively) in comparison to untreated/lipofectamine-treated controls. Treatment with miR-708-5p inhibitor elicited no significant neuritogenic effect (20 ± 6.2 RGC with neurites) compared to untreated/lipofectamine-treated controls.

## 4. Discussion

The present study identified key miRNA expressed within RGC. From these miRNA, a profile was established detailing those miRNA whose expression is significantly perturbed under traumatic and/or degenerative damage to the optic nerve. From this list, we selected individual candidates and, through miRNA modulation in retinal culture, showed that some of the miRNA are causative in the degenerative response, thus representing ideal targets for future therapies.

Previous studies have identified key miRNA changes in glaucomatous retina, but as the RGC make up less than 1% of the total retinal cell population [28], it is understandable that these changes may not reflect RGC miRNA changes. In one such study, nine miRNA were identified as differentially expressed in glaucomatous rodent retina in comparison to healthy retina [14]. This substantially smaller number of candidates compared to our own study likely reflects the challenge of detecting changes in RGC while analysing total retina. Jayaram and colleagues found eight miRNA (miR-16, -29b, -106b, -497, -181c, -25, -204, and let-7a) to be downregulated and miR-27a upregulated. Of the downregulated miRNA, miR-16, 106b, -497 and let-7a are pro-apoptotic whereas the upregulated miR-27a is anti-apoptotic. Authors suggested this reflected a shift to a pro-apoptotic phenotype which ultimately fails. Our data, which looks specifically at RGC miRNA, however, tells a different story. In contrast, we found the pro-apoptotic miR-16 to drastically increase in glaucomatous RGC, as did let-7a, which reflects the apoptosis ongoing within these RGC. We did, however, see congruence between our data and the authors with regard to miR-204, -181c, -25, and -27a. Regarding miR-181, the changes were so substantial that we tested this miRNA further (discussed below). We did not detect any significant changes regarding miR-497, -29b or -106b.

To test candidate miRNA for their causative role in neurodegeneration, we modulated their abundance in heterogenous retinal cultures to counteract changes we saw in our miRNAseq. Previous research has demonstrated that miRNA-181a negatively regulates MAPK/ERK signalling, leading to a reduction in RhoA [29]. RhoA is a GTPase protein that initiates actin destabilization and axonal growth cone collapse [1]. RhoA inhibitors have been suggested as therapy to promote axon regeneration [30] and miR-181a specifically has been shown to be involved in axonal growth and connectivity in the RGC of the developing visual system [29]. Based on the above findings, it is thus no surprise that significant reductions in miR-181 of RGC occur after injury, as these injuries are characterized by axonal degeneration, which would require disinhibition of the miR-181-RhoA axis. Delivery of miR-181, therefore, initiates an axon regenerative effect on cultured RGC, presumably through inhibition of RhoA and subsequent growth cone collapse. Note that studies suggest the 5p strand of miR-181 is the dominant strand in tissues, whereas we found that for miR-181a, the 3p strand was most abundant in RGC [31]. miR-181 exist as four subtypes: miR-181a,b,c, and d. These likely have the same functions and targets given the substantial similarity of their sequences, yet differences are seen in where they are expressed. miR-181a and b are more prevalent in neurons while miR-181c and d expression is more widespread.

Interestingly, while we observed a neuritogenic effect, there was no such survival effect and even a trend towards RGC loss. Several studies have shown that neuroprotection can be achieved through the downregulation of miR-181 including cortical neurons in a model of stroke [32], and RGC in a model of Leber’s hereditary optic neuropathy [33].

We also observed significant neuroprotective effects after inhibition of miR-194-5p. Few studies have been published on the role of miR-194-5p in CNS injury; however, one study corroborates our findings. miR-194-5p delivery into primary cortical neurons, while not directly causative of neuronal cell death, did mediate neuronal death instigated via oxygen-glucose deprivation [34]. Cortical neurons with enhanced levels of miR-194-5p were exquisitely sensitive to this form of damage. Deep hypothermic circulatory arrest has been a suggested treatment for stroke and, in patients receiving this treatment, miR-194-5p levels are significantly reduced, and this association has been confirmed in cortical cultures.

Modulation of miR-708-5p elicited no significant effects within our in vitro RGC culture model. Whether this highlights that miR-708-5p is not causative in the degenerative/regenerative pathways of the CNS, or this merely reflects a limitation in our in vitro model, is unknown. miR-708-5p research has focused predominantly on cancer, and on its paradoxically oncogenic and tumor-suppressive roles [35]. It is not uncommon for molecules involved in cancer to also be targets for axogenic therapies, given their mutual reliance on cell growth and protein synthesis, as has been seen for PTEN [36].

The final miRNA we targeted was miR-664 (referred to as miR-664-2 in rat) and downregulation using an inhibitor elicited a significant neuroprotective effect. Little is known regarding this miRNA in the context of neuronal degeneration, making any discussion regarding its mechanism here speculative. One study found that miR-664 induces neuronal differentiation of SH-SY5Y cells, and when analysing the downstream mRNA that were targeted, identified five predicted gene targets: INSM, FLT1, FAT3, EDIL3 and DCX [37]. A further two targets that have been demonstrated to have functional consequences are serum response factor and Wnt1 [38].

Modulating miRNA is not a novel treatment for RGC and several studies have been published showing their efficacy. In a mouse model of glaucoma, downregulation of miR-149 led to a significant increase in RGC numbers while minimizing their ultrastructural alterations [39]. In a second glaucoma model, utilizing NMDA, overexpression of miR-93-5p significantly attenuated the reduced viability of these cells in this model [40]. The authors found that this therapeutic effect was achieved through the targeting of miR-93-5p to PTEN. Interestingly, neither miR-149 or miR-93-5p were one of the candidates we identified in our miRNAseq, potentially suggesting that these miRNA are acting on intermediary cells such as glia.

An important observation of this study is of the differences between TON in the acute (2 days) and more chronic (7 days) phase, as well as the differences between glaucoma and TON. We observed significant differences between 2 and 7 days post-ONC. While Group 4 (7-day) demonstrated significant changes from healthy retina, Group 3 (2-day) yielded comparatively few changes. These data strongly suggest that miRNA changes are not immediate and represent the more chronic response to CNS injury. Comparing Group 2 (glaucoma) and 4 (7-days post-ONC) also yields significant differences, yet these are less numerous than the comparisons between Group 3 and 4. While glaucoma and TON have surface level similarities (RGC death resulting from optic nerve damage), they are distinct. The biggest difference is that TON is associated with axotomy (and subsequent abortive regeneration), whereas glaucoma is strictly RGC loss. It can thus be speculated that the similarities between Group 2 and 4 represent miRNA associated with RGC death, whereas miRNA changes that were different between Group 2 and 4 are those associated with the abortive axon regenerative process. The development of new neuroprotective treatments would, thus, presumably target these miRNA with overlapping changes, whereas pro-axogenic therapies would target miRNA that change in Group 2 but not Group 4.

An important consideration is that “RGC” is an umbrella term that includes over 40 subtypes [13]. The purification process used in this study relies on the surface expression of Thy1, the most ubiquitous surface marker for the RGC subtypes that excludes other retinal cells. Thy1 is still, however, only found on approximately 80% of fluorogold-labelled RGC [41] and 82% of RBPMS-labelled RGC [42] The remaining 20% of RGC were, thus, presumably not purified and not sequenced in this study. Until suitable surface markers or methods for purifying the entire population are discovered, sequencing of 80% of the RGC population is the best that can be achieved.

A second consideration is the optimal method by which miRNA are to be delivered into RGC. In this study, we used lipofectamine to deliver miRNA mimics/AntagomiRs into RGC, an approach feasible in vitro but less so in vivo. Alternative approaches would need to be considered, and include the use of viral vectors [43] or exosome/small extracellular vesicle carriers [44].

In conclusion, we have determined the miRNA changes that occur after both traumatic damage to the optic nerve (TON), and degenerative damage (glaucoma/ocular hypertension). We have also confirmed that at least some of these miRNA are not just consequences of the damage, but also may play a causative role, thus acting at novel targets for treatment. Further testing of selected miRNA in animal models and characterization of their targets in the retina are the next important steps in the search for new neuroprotective strategies.

## Figures and Tables

**Figure 1 cells-10-01564-f001:**
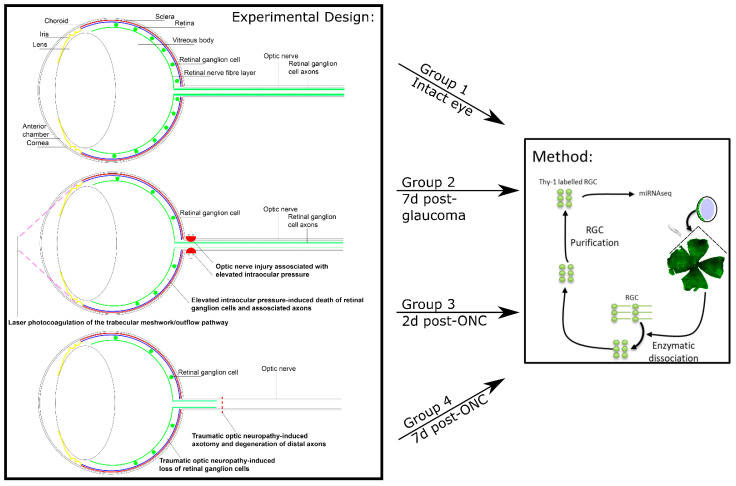
The experimental design of the present study. Timeline and groupings of the present study, detailing the models used and the method of RGC purification.

**Figure 2 cells-10-01564-f002:**
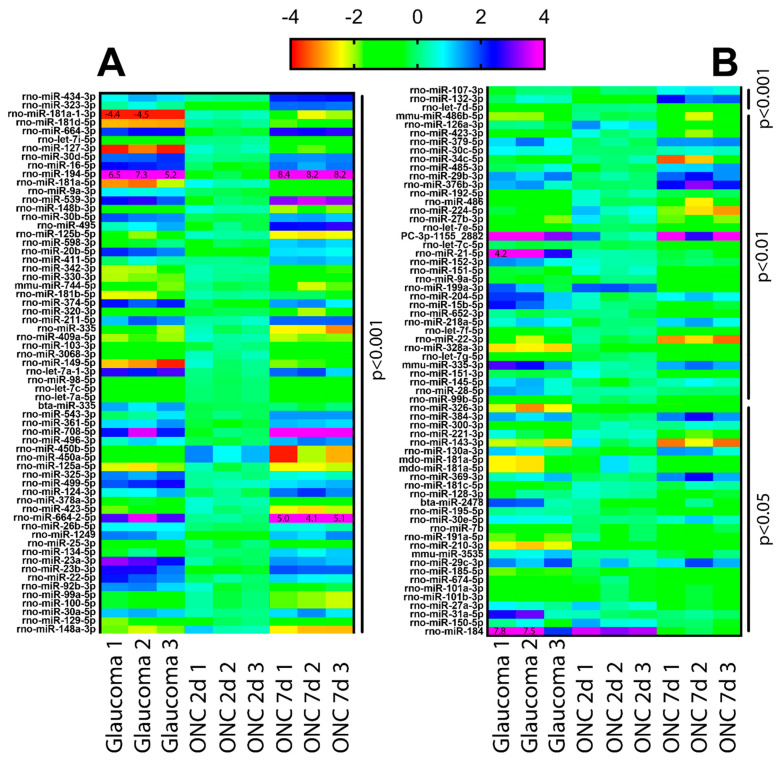
miRNA fold change heat map profile. The miRNA profile of purified RGC in diseased retina, displayed as a heat map of the log_2_-fold change in comparison with RGC from healthy/uninjured retina. Red/yellow indicates a lower abundance in comparison to healthy RGC whereas blue/purple indicate a comparatively higher abundance. The 14 values that exceed the scale are labelled directly. Only miRNA that are significantly different from their abundance in healthy RGC are shown, with the *p* values (*p* < 0.001 in Panel **A**; *p* < 0.01/0.05 in Panel **B**) given to the right of the heat maps.

**Figure 3 cells-10-01564-f003:**
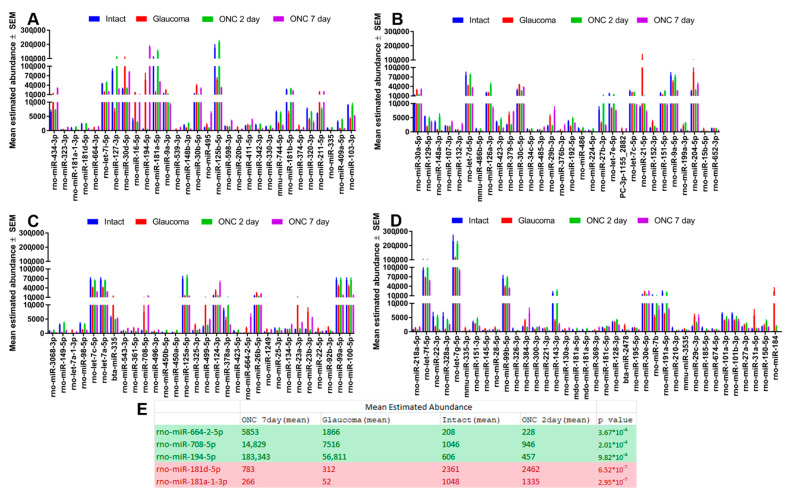
miRNA abundance profile. Histograms displaying the average abundance values for the miRNA detailed in Figure 2. (miRNA significantly different between the RGC of healthy and diseased retina). miRNA are displayed in the same order as in Figure 2, starting with those that are the most significantly different (**A**–**D**). The five candidate miRNA that were further tested for their neuroprotective efficacy are included in the table (**E**) along with their respective abundance counts. Those in green demonstrated an increase in abundance after injury, whereas those in red demonstrated a decrease.

**Figure 4 cells-10-01564-f004:**
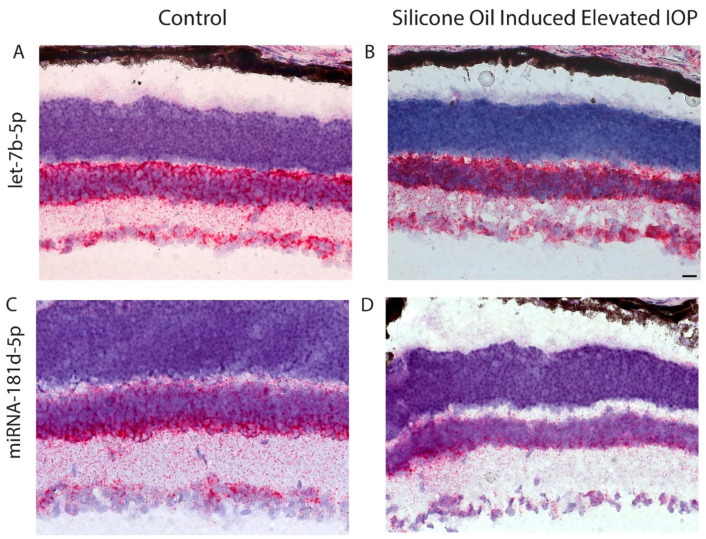
Expression of miRNA in RGC and changes after elevation of IOP. In situ hybridization shows Let-7b-5p expression in the INL and GCL (**A**). No change in signal was detected in Let-7b-5p after elevated IOP (**B**). Expression of miRNA-181d-5p is localized to the INL and GCL (**C**). Following elevation of IOP, the signal of miRNA-181d-5p is appreciably lower in both the GCL and INL (**D**). Scale bar: 10 µm.

**Figure 5 cells-10-01564-f005:**
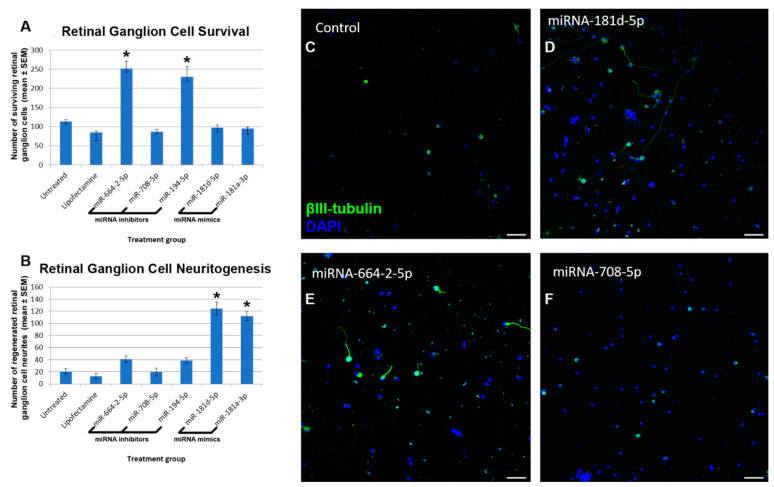
Effects of miRNA modulation on RGC neuroprotection and neuritogenesis. The number of RGC (**A**) and the number of RGC bearing neurites (**B**) in heterogenous retinal cultures with and without miRNA modulation is shown. Asterisks indicate significant differences from both untreated and lipofectamine only controls (*p* < 0.05). miR-181a and miR181d refer to the use of miRNA mimics whereas miR-194, miR-708, and miR664-2 refer to the use of mirVana miRNA inhibitors. Representative images of heterogenous retinal cultures treated with: lipofectamine only (**C**), miR-181d mimic (**D**), miR-664-2 inhibitor (**E**), or miR-708 inhibitor (**F**). All images are representative of the entire culture, nine separate culture wells/treatment, with every 3 wells using a different animal (scale bars: 50 µm). Sections were stained for βIII-tubulin (green) and DAPI (blue).

**Table 1 cells-10-01564-t001:** The most abundant miRNA identified in retinal ganglion cells, Muller glia, neural retina, CA3 hippocampus, and cortex. The miRNA labelled in red are those unique to RGC and not found to be abundant in the other cell types.

Retinal Ganglion Cells	Muller Glia *Wohl and Reh* (2016)	Neural Retina without Muller Glia *Wohl and Reh* (2016)	CA3 Hippocampus *Shinohara* et al. (2011)	Cortex *Yao* et al. (2012)
let-7g	mir-204	mir-124	let-7c	mir-128
mir-182	mir-125b-5p	mir-183	let-7a	let-7c
mir-125-5p	mir-9	mir-96	let-7f	let-7a
mir-181a-5p	mir-181a	mir-181a	let-7b	let-7f
let-7f	let-7c	let-7g	mir-9	let-7d
mir-29a-3p	mir-720	mir-1944	mir-138	mir-29a
mir-127-3p	mir-99a	mir-16	mir-30e	let-7b
let-7d	let-7b	let-7d	mir-126	mir-124
mir-24a-5p	let-7g	mir-30c	mir-24	mir-103
mir-9a-5p	mic-30c	mir-29c	mir-143	let-7e
mir-125a-5p	let-7d	mir-720	mir-21	mir-107
let-7c-5p	mir-135a	let-7b	mir-127	mir-99a
let-7a	mir-29a	mir-204	mir-30a	let-7i
mir-99a-5p	let-7a	let-7a	mir-26a	mir-30d
mir-100-5p	mir-16	let-7c	mir-9 *	mir-9
mir-99b-5p	mir-125a-5p	mir-211	mir-29b	mir-30a
let-7i	mir-100	mir-125b-5p	mir-103	mir-101a
mir-183-5p	mir-1944	mir-129-3p	mir-27b	mir-219-2-3p
mir-7a-5p	mir-22	mir-9	mir-101a	mir-185
mir-96-5p	mir-30d	mir-25	mir-30d	
mir-30d-5p	let-7e	mir-22	mir-181b	
mir-30c-5p	mir-335-5p	mir-342-3p	mir-379	
mir-181b-5p	mir-23a	mir-125a-5p	mir-101b	
mir-204-5p		let-7f	mir-132	
let-7c-5p		mir-151-5p	mir-125b-5p	
let-7b-5p		mir-1186	mir-218	
mir-191a-5p		let-7e	mir-411	
mir-126a-3p		mir-182	mir-378	
mir-143-3p		mir-451	let-7i	
mir-151-5p		mir-29a	mir-124	
let-7e-5p		mir-30d	mir-146b	
mir-124-3p		mir-210	let-7d	
mir-26b-5p		let-7i		
mir-9a-3p		mir-500		
mir-30e-5p		mir-15b		
mir-30a-5p		mir-1224		
mir-30b-5p		mir-301a		
mir-7b		mir-1937a + b		
mir-129-2-3p		mir-15a		

## Data Availability

All data relevant to the present study can be found within the paper, or as Appendix A.

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
