# Peer review of "miRNA Changes in Retinal Ganglion Cells after Optic Nerve Crush and Glaucomatous Damage"

_cells, 2021, doi:10.3390/cells10071564_

Round 1
Reviewer 1 Report
The authors try to identify the role of miRNA changes in retinal ganglion cells after optic nerve crush and glaucomatous damage. Even though it is an interesting study the significance and novelty of contribution are much less. The authors have not done crucial experiments to support their findings.
(i) The authors have used thy1 as the selective marker to choose RGCs and publicly available single-cell retina data shows that thy1 is also present in certain Amacrine cell types to some extent. The authors can not rule out the possibility that there may be also other cell types in their RGC enrichment. Since the authors claim that this study is novel in showing miRNAs exclusive to RGC, the presence of other cell types has to be discussed clearly. Since their data is based on cell culture it is important to show the purity of the culture. Add the IF staining data for RBPMS at least as a supplementary.
(ii) Since (mir-7a-5p and mir-191a-5p) are specifically enriched in RGCs, the authors need to show their specific presence by IF or IHC. The miRNAs you showed miRNA-181d-5p is highly expressed in INL and hence do not justify the significance of the paper that claims RGC specific miRNA expression changes.
(iii) Since expression of miRNA-181d-5p significantly decreased in glaucoma, try to show the IF between control and glaucoma tissue to show the difference of expression.
(iv) Highlight the miRNAs in Fig.2 AND 3 that have been used in follow-up studies. Use color codes to differentiate miRNAs used in different experiments, to make it easy for readers to understand. Also when using inhibitors or mimics, mark specifically with a letter (M or I) in the Figure labels.
(v) Kindly add the images for all the mimics and inhibitors used as you have done the data already. Based on the results miR-181d-5p did not show a significant cell survival rate, whereas the Fig.5D panel shows more DAPI cells than the control similar to miR-664-2-5p. The images and description do not correlate. The panel labeling is missing.
Reviewer 2 Report
Summary.
This study characterized changes of the miRNA profile of RGCs isolated from retinas of established rat models of optic nerve crush (ONC) and laser-induced ocular hypertension (OHT). Previous studies have identified significant miRNA changes in the total retinal cell population of glaucomatous retina, but these were not specific for the relatively small (1%) RGC population. The study identified more than 100 miRNA changes compared to controls retinas 7 days after ONC and OHT, whereas little miRNA changes occurred 2 day after ONC, which would suggest that miRNA changes represent a chronic response to CNS injury. Investigation of candidate miRNAs in heterogeneous RGC cultures using mimics/AntagomiR showed that miR-194 inhibitor and miR-664-2 inhibitor were neuroprotective, whereas miR-181a mimic and miR-181d-5p mimic elicited significant RGC neuritogenesis. The study suggested that some of altered miRNA may play a causative role in traumatic optic neuropathy. Altered miRNAs may represent novel targets for treatment, which would imply suitable in vivo approaches such as the use of viral vectors or exosome/small extracellular vesicle carriers.
Main comment
This an interesting and well written study with solid results, significant translational relevance, and the intriguing speculation that similarities and differences of miRNA profiles between OHT and ONC may reflect processes associated with either RGC death or abortive axon regeneration. I do not have major comments, but I have a couple minor comments the authors may consider.
Figure 5. The legend describes A-F panels that are not labeled in the figure. Also, the legend mentions significance asterisks that are not present in the bar graphs.
Reviewer 3 Report
The authors described the miRNA profile in RGCs after TON and Ocular hypertension induction, and it was identified 5 miRNA that demonstrated major deviations from healthy retinas.
However, the manuscript requires some considerations:
- Line 148 – the authors reported that it was prepared retinal cultures from retinal cell suspension, however, it is not clear for what it was used. It is described that RGCs were purified from retinal cell suspension by magnetic beads, and the purified RGCs were used for RNAseq and a portion of RGCs were plate for the experiments regarding figure 5. However, for retinal cultures described in line 148 there is no further experiments described.
- Regarding purified RGCs that were plated for the experiments regarding figure 5, it was not described the medium used, the time in culture and the density to plate the cells. The authors only described that “A portion of RGC were plated at a density 5000 RGC/well to confirm RGC purity”. Please add this information.
- Line 184 – In this section (2.10) of Materials and Methods it is missing information about the markers used for RGC countings and neurite number coutings (results presented in figure 5). Moreover, the authors did not describe how do they acquire the images as well as how do they counted both RGC and neurite number. This information should be included.
- Line 196 – The authors reported that “graphs constructed using Graphpad Prism 7.01”, however, it seems that the graphs presented in figure 5 are from excel. Please correct this information.
- Line 224 – The graphs in figure 3 is not easy to follow, and it not easy to find differences that the authors are reporting in the text. At least the authors should present individuals graphs for each miRNA that was selected to proceed in the study, and clearly describe these effects in the results section (the authors only describe the results for miR-195a-5 and miR-181).
- It is possible to observed in figure 2, at least, 2 others miRNA that are decreased (miR-149-5p and miR-127-3p). Why was this not considered to proceed in the study?
- Line 251 – The authors reported that “Treatment with miR-664-2-5p or miR-194-5p inhibitors elicited a trend towards neuritogenesis”. Even though it is not statistical significant, the authors must report the exact p value in order to demonstrate the trend of the effect.
- Line 283 – “In contrast, we found the pro-apoptotic miR-16 to drastically increase in glaucomatous RGC”. In this part of the discussion, it is reported an increase in a miRNA (miR-16) which is a result that is not presented in the results sections of the manuscript. In my point of view, it is missing a general description of the main miRNA that are changed, probably it can be divided into categories in order to give a big picture of which are the main categories changed in each model (like pro-apoptotic, anti-apoptotic, axonal growth…).
Considerations about the figures:
Figure 1 – Very well schematic representation of the experimental design of the study. However, I suggest to the authors to insert information about eye structures (like: cornea, iris, lens, sclera…) only in the first schematic eye representation (intact eye). That way, the other 2 eyes representations will be more clean and it will be better to understand the effect of the models used in this study.
Figure 5 – In the figure legend there is different description about figure 5A, 5B… however, the letters A, B, C… are missing in the figure. Moreover, instead of putting only the identification of the miRNA that was manipulated, is more visual if the authors include information in the graphs/figure if it was used miRNA mimics or inhibitors. Moreover, the authors should include representative images for all miRNA that was manipulated. As an example, the authors report that miR-194-5p also increased RGCs survival, that way the representative image for these results can be included. Please include the statistical differences in the graphs. The authors indicate that “Asterisks indicate significant differences from both untreated and lipofectamine only controls (p < 0.05)”, but the asterisks are missing in the graphs.
Round 2
Reviewer 1 Report
The paper can be accepted